# Analysis of Tick Surface Decontamination Methods

**DOI:** 10.3390/microorganisms8070987

**Published:** 2020-06-30

**Authors:** Angeline Hoffmann, Volker Fingerle, Matthias Noll

**Affiliations:** 1Institute for Bioanalysis, Department of Applied Sciences, Coburg University of Applied Sciences and Arts, 96450 Coburg, Germany; angeline.hoffmann@hs-coburg.de; 2Bavarian Health and Food Safety Authority (LGL), National Reference Center for Borrelia, 85764 Oberschleißheim, Germany; volker.fingerle@lgl.bayern.de

**Keywords:** amplicon sequencing, bacterial 16S rRNA gene, ribosomal RNA, surface decontamination, ticks

## Abstract

Various microbial pathogens have been found in ticks such as *Ixodes ricinus*. However, most studies assessed tick microbiomes without prior decontamination of the tick surface, which may alter the results and mislead conclusions regarding the composition of the tick-borne microbiome. The aim of this study was to test four different decontamination methods, namely (i.) 70% ethanol, (ii.) DNA Away, (iii.) 5% sodium hypochlorite and (iv.) Reactive Skin Decontamination Lotion (RSDL), which have been previously reported for tick surface and animal or human skin decontamination. To test the efficiency of decontamination, we contaminated each tick with a defined mixture of *Escherichia coli*, *Micrococcus luteus*, *Pseudomonas fluorescens*, dog saliva and human sweat. No contamination was used as a negative control, and for a positive control, a no decontamination strategy was carried out. After nucleic acid extraction, the recovery rate of contaminants was determined for RNA and DNA samples by qPCR and tick-borne microbiome analyses by bacterial 16S rRNA and 16S rRNA gene amplicon sequencing. Ticks treated with 5% sodium hypochlorite revealed the lowest number of contaminants followed by DNA Away, RSDL and 70% ethanol. Moreover, tick microbiomes after 5% sodium hypochlorite decontamination clustered with negative controls. Therefore, the efficiency of decontamination was optimal with 5% sodium hypochlorite and is recommended for upcoming studies to address the unbiased detection of tick-borne pathogens.

## 1. Introduction

Ticks are widespread obligate bloodsucking ectoparasites with diverse wildlife, livestock, domestic animals and humans as hosts [1]. Ticks serve as vectors for many tick-borne pathogens (TBP), which can be transmitted from and/or to its hosts by each blood meal [2]. *Ixodes ricinus* is the most common tick species in Germany and the vector for a vast range of TBP, including *Borrelia*, *Rickettsia* or *Coxiella* that can cause Lyme disease, rickettsial disease or Q fever [3,4,5]. The diversity and quantity of TBPs have been assessed by a variety of molecular methodologies, including next-generation sequencing (NGS) such as amplicon sequencing [1]. NGS is a quick and cost-effective technology that enhanced our knowledge of genes and genomes of single cells and compositions of complex microbial communities in the last decade [1,6]. The 16S ribosomal RNA (rRNA) and its gene have been frequently used as a target for amplicon sequencing to identify bacterial taxa [1,6,7]. Therefore, nine hypervariable regions (V1–V9) in the rRNA gene have been assessed of which regions V1–V4 are most commonly used to explore the bacterial sequence diversity in ticks [8,9,10].

TBPs are part of a diverse tick microbiome consisting of varying bacterial diversities that are unevenly distributed in the tick organs [1,10,11,12]. The diversity and quantity of the components of the internal tick microbiome are essential for the tick life cycle, including fitness, survival and immunity [11]. The external tick microbiome is most likely transferred or affected either through tick–host interaction during blood-meals or by accompanying microorganisms from the tick habitat. Nevertheless, the external tick microbiome and its influence is under-investigated and both have to be included more extensively in microbiome studies [13]. However, the external microbiome of other arthropods and insects was investigated, and such cuticular microbiomes were essential to protect the vector from environmental stressors and/or useful for host recognition [14,15,16].

In contrast, the microbial diversity of the external human microbiome is well known and comprises approximately 1 × 10^6^ bacteria per cm^2^ [17]. Moreover, human skin decontamination for the aseptic treatment of skin lesions prior to vaccinations or surgeries is common to reduce the impact of the external skin microbiome as a potential source of infection [18]. Lotions, including povidone-iodine, chlorhexidine, ethanol and reactive skin decontamination lotion (RSDL), are frequently used for skin surface decontamination [19,20].

The skin host microbiome (such as that of mammals like humans or dogs) is potentially part of the external tick microbiome, which can contaminate the entire tick microbiome’s analyses. Such contaminations may cause misleading detection of pathogens with therapeutic implications. Our enquiry revealed that out of 30 several studies addressing the internal tick microbiome, only 11 studies had included a decontamination assay. Greay et al. reported that the decontamination of a tick surface microbiome is heterogeneously carried out and the majority decontaminated by washing or rinsing with 70% or 100% ethanol treatment [1,12,21,22], while only a few studies used bleaching solutions such as sodium hypochlorite [22,23,24,25]. Ethanol, sodium hypochlorite or DNA away are frequently used for surface decontamination [26,27] and each can cause a degradation of nucleic acids. RSDL is a promising decontamination method for tick surface microbiomes as its compounds have been optimised to eliminate the attack of toxic chemicals and biologicals of human skin surface [28,29]. However, detailed information of decontamination procedures or protocols for the tick surface are not publicly available. If nucleic-acid-based analyses of the tick microbiome are planned, the efficient removal of nucleic acids from microbial contaminants is needed. The efficiency of nucleic acid extraction procedures has been reviewed frequently [30]. Gram-negative (e.g., *Escherichia coli* and *Pseudomonas fluorescens*) and Gram-positive bacterial strains (e.g., *Micrococcus luteus*) were spiked to different environmental samples (e.g., soil) prior to nucleic acid extraction to evaluate the nucleic-acid-based recovery rate of each strain after extraction. As the chemical rupture of Gram-negative bacterial cell walls is more likely than of Gram-positive bacteria [30], the spiking of such bacteria is also relevant to test the efficiency of skin decontamination methods. Both DNA- and RNA-based pathogens in ticks are of diagnostic interest. Thus, the removal efficiency of the external tick microbiome for DNA and RNA samples should be addressed.

This study aimed to test the efficiency of four different decontamination methods, namely 70% ethanol, DNA Away, 5% sodium hypochlorite and the RSDL of an artificially contaminated tick surface. The contamination procedures were carried out as described previously on human skin [31]. Efficiency was quantitatively and qualitatively evaluated for RNA (reverse transcribed to complementary DNA) and DNA samples by qPCR and bacterial 16S rRNA gene amplicon sequencing.

## 2. Materials and Methods

### 2.1. Ethics Statement

Permission for tick collection at the study site was requested at the government of Lower Franconia at Würzburg, workspace animal protection and animal testing. For this examination, no formal permit was required.

### 2.2. Tick Sample Collection

A total of 62 host-seeking individual specimens of adult *I. ricinus* were collected in September 2018 at the Hofgarten Coburg (50°15′39″, 10°58′24″). All ticks were collected by flagging a 1-m^2^ white cloth, as explained earlier [32]. Each tick was transferred with a tweezer, which was freshly disinfected after each tick, to a 1.5-mL sterile reaction tube. The tubes were immediately transported to the lab and stored for 16 h at 4 °C until further processing.

### 2.3. Contamination and Decontamination of Ticks

The bacterial strains *E. coli* (DSM 423), *P. fluorescens* (DSM 4358) and *M. luteus* (DSM 20030) were chosen based on different cell wall constitutions for artificial spiking to the external tick microbiome to evaluate the efficiency of decontamination methods. So far, only members of the genus *Pseudomonas* were found in tick microbiomes [33], while *Escherichia-Shigella* or *Microcococcus* were not reported [34,35,36,37,38]. In addition, dog saliva and human sweat were artificially spiked to the external tick microbiome to add complex microbial communities of potential hosts. The microbiome of dog saliva and human sweat has been frequently reported and the most abundant bacterial genera of its compositions were not found in tick microbiomes [39,40], which is also true for this study (Appendix A).

Ticks were randomly subdivided into nine treatments (see Table 1). Except for ticks of the negative control (DKA 1 NC, DKA 2 NC, DKA 3 NC, DKA 4 NC; *n* = 12), each tick (*n* = 50) was separately contaminated by placing them [22] in a 1.5-mL tube for 5 min at room temperature containing 45 μL of a defined contamination solution of *M. luteus*, *P. fluorescens*, *E. coli*, dog salvia and human sweat (see Table 2). Contamination solution was discarded thereafter and contaminated ticks, except for ticks of the positive control (PC; *n* = 10), were decontaminated by placing them for 5 min at room temperature in a 1.5-mL tube containing 50 µL of (i.) 70% ethanol, (ii.) DNA Away, (iii.) 5% sodium hypochlorite or (iv.) RSDL (see Table 1). A decontamination time of 5 min has been used for a sodium hypochlorite treatment previously [41]. Afterwards, the decontamination solution was discarded, and the remaining solution was completely evaporated by SpeedVac.

### 2.4. Nucleic Acid Extraction

Nucleic acids were extracted from each treated tick (Table 1) by chemagic™ Viral DNA/RNA kit (PerkinElmer chemagen Technologie GmbH, Baesweiler, Germany) on the chemagic™ MSM I instrument (PerkinElmer chemagen Technologie GmbH) as specified by the manufacturer. Briefly, each tick was homogenized by adding 50 µL of isotonic saline solution (0.9% 154 mM NaCl; pH 5.7) and one 5-mm steel bead (Qiagen GmbH, Hilden, Germany) by a TissueLyser II (Qiagen GmbH) for 4 min. After that, each homogenized tick was lysed by a lysis buffer of the chemagic™ Viral DNA/RNA kit (including protease and Poly (A) RNA), and nucleic acids were automatically extracted by using magnetic beads as explained by the manufacturer. Nucleic acid extractions were carried out at SYNLAB Holding GmbH (Weiden in der Oberpfalz, Germany). Nucleic acid extract from each tick was equally divided into two parts, and one part was used for genomic DNA analyses, while the other part was used for RNA analyses. The latter was treated with DNase for 30 min at 37 °C (RQ1 RNase-Free DNase, Promega) and the success of DNA degradation was checked by PCR, as explained earlier [42]. Afterwards, RNA was transcribed into cDNA by a high capacity cDNA reverse transcription kit (Thermo Fisher Scientific Inc., Applied Biosystems™, Waltham, MA, USA), as specified by the manufacturer.

### 2.5. Quantitative PCR Amplification for DNA and cDNA Samples

The qPCR for DNA and cDNA samples of each nucleic acid extract was performed to quantify the contaminants *E. coli* (primers: 395F: 5′-CATGCCGCGTGTATGAAGAA-3′ and 470R: 5′-CGGGTAACGTCAATGAGCAAA-3′ [43]), *P. fluorescens* (primers: 433F: 5′-CTGACACCAAGGCTATCG-3′ and 576R: 5′-GCCTTCTACAACCGACAG-3′ [44]), and *M. luteus* (primers: 172F: 5′-AACCGTTAGACTCCGAGCAC-3′ and 393R: 5′-CAGGAGCGTATTGCCGATGA-3′, this study). Primer pairs were evaluated as outlined previously [45]. Twofold concentrated iTaq Universal SYBR Green Supermix (Bio-Rad Laboratories GmbH, Feldkirchen, Germany), 1 μL of template (1:1, 1:10 and 1:100 dilution in three replicates) or nuclease-free master mix were run as a negative control for qPCR in a final volume of 20 µL. CFX96™ Real-Time System C1000™ Thermal Cycler (Bio-Rad Laboratories GmbH) was used for qPCR with the following thermal conditions: initial denaturation at 95 °C for 3 min for DNA samples or 30 s for cDNA samples followed by 35 cycles of denaturation at 95 °C (5 s), annealing at 48 °C for *E. coli*, 46 °C for *P. fluorescens* and 50 °C for *M. luteus* (30 s), extension 60 °C (30 s) and the final elongation at 72 °C (10 min).

To calculate the gene copy numbers, the initial cell number of *E. coli*, *M. luteus* and *P. fluorescens* were microscopically estimated in a Neubauer counting chamber (Carl Roth GmbH & Co KG, Karlsruhe, Germany) followed by a nucleic acid extraction as explained above. Thereafter, a quantity of genomic DNA (gDNA) of each strain was used as a standard to correlate the PCR-cycle threshold values of nucleic acid extracts of each sample to respective gene copy numbers. The gDNA concentration per PCR reaction of *E. coli*, *M. luteus* and *P. fluorescens* standard ranged from 6 × 10^9^ to 6 × 10^1^, 8 × 10^8^ to 8 × 10^4^ and 2 × 10^8^ to 2 × 10^2^ gene copies, respectively. Multiple dilutions were run simultaneously to check for inhibitors in qPCR assays. Based on these results, non-diluted DNA and cDNA extracts were best suited for qPCR analyses (data not shown). Cycle threshold and efficiency were calculated by the Bio-Rad software CFX manager version 3.1.

### 2.6. Bacterial 16S rRNA Gene Sequencing for DNA and cDNA Samples

To create amplicon sequencing libraries, the V3–V4 region of the bacterial 16S rRNA gene was amplified with the primer set (341F: 5′-CCTACGGGNGGCWGCAG-3′ and 785R: 5′-GACTACHVGGGTATCTAATCC-3′ [7]) of each of the 62 genomic DNA and 62 cDNA extracts. Amplicon sequence libraries were made by adding inline barcodes and Illumina sequencing adapters using the Nextera XT Index Kit (Illumina, San Diego, CA, USA) and MiSeq Reagent Kit v3 600 cycles (Illumina, San Diego, CA, USA) according to manufacturer’s instructions. PCR products for library preparation were purified by AMPure XP beads (Beckman Coulter, Brea, CA, USA) and 5 μL of DNA or cDNA was equimolar pooled for each library (up to 96 libraries) with unique indices for each tick and treatment (Table 1). The sequencing of libraries was performed by 300 bp paired-end sequencing on an Illumina MiSeq platform (Illumina MiSeq V3; Illumina) based on a standard protocol from the manufacturer. Amplicon sequencing and a basic sequence quality check were carried out by LGC Genomics GmbH (Berlin, Germany).

### 2.7. Bioinformatics

Raw data pre-processing with demultiplexing, sorting, adapter trimming and the merging of reads were assembled using the Illumina bcl2fastq conversion software v2.20 and BBMerge [46]. Afterwards, the sequence quality of the reads was checked with the FastQC software, version 0.11.8 [47]. Sequence pre-processing was carried out separately for DNA and for cDNA samples as described by Buettner and Noll (2018) with minor modifications [48]. Sequence pre-processing and Operational Taxonomic Units (OTUs) picking from amplicons was carried out by using Mothur 1.35.1 [49]. Sequences with an average Phred quality score over 33 were aligned against the 16S Mothur-Silva SEED r119 reference alignment [50]. Short alignments were filtered, and sequencing errors were reduced by pre-clustering, where a maximum of one nucleotide mismatch per 100 nucleotides in a cluster was allowed. Singletons and chimeras were eliminated, the latter with the UCHIME algorithm [51]. For picking OTUs, sequences were classified taxonomically against the Silvia references classification and were thereafter removed from other domains of life. Using the cluster.split method, OTUs were picked and assigned to a taxonomic level by clustering at the 97% identity level [52], leading to OTU tables for DNA and cDNA samples.

### 2.8. Statistics

Rarefaction analysis, the estimation of alpha diversity (OTU richness, Shannon index, Pielou’s Evenness) and OTU richness estimators (Chao1 and an abundance-based coverage estimator (ACE)) were performed for DNA and cDNA samples in RStudio and the packages vegan 2.5-4. [53,54]. Correspondence analyses were performed with transformed bacterial OTU matrices (taxonomically summarized on genus level and additionally summarized on decontamination strategies) using FactorMineR [55]. Cluster analyses were carried out with a Euclidian distance according to the ward.D2 method between the composition of tick microbiomes (first two dimensions of correspondence analysis) for DNA and cDNA samples by using the functions dist and hclust in the package stats [56] and visualized by the package dendextend [57]. The Bray Curtis similarity heatmap and cluster analysis of OTU matrices were calculated with the packages, vegan 2.5-4. and gplots 3.0.3. [53,58]. Relative OTU abundances were calculated for each tick extract, as explained earlier [42]. Thereafter, OTUs were taxonomically summarized on a genus level (Appendix A) and the ten most abundant genera were visualized with Origin 2017 (OriginLab Corporation, Northampton, MA, USA). Significant effects (*p* < 0.05) between bacterial OTUs, gene copy numbers, DNA and cDNA samples and respective decontamination treatment were calculated by one-way ANOVA with a post hoc adjusted Tukey test in Origin 2017 (OriginLab Corporation).

### 2.9. Nucleotide Sequence Accession Numbers

The bacterial 16S rRNA gene sequences for DNA and cDNA samples were deposited in the NCBI nucleotide sequence databases under accession PRJNA631133.

## 3. Results

### 3.1. Reduction of Artificial Bacterial Contaminants

A decontamination treatment with 5% sodium hypochlorite (DKA 3) was most efficient, followed by DNA Away (DKA 2), RSDL (DKA 4) and 70% ethanol (DKA 1) for DNA samples (Figure 1A). Regardless of the respective decontamination treatment, *M. luteus* was significantly less efficiently removed compared to *P. fluorescens* (*p* = 2.34 × 10^−10^) and *E. coli* (*p* = 7.79 × 10^−12^) for DNA samples (Figure 1A). Efficiency in removing *P*. fluorescens and *E. coli* was similar to each other irrespectively of decontamination treatment (*p* = 0.650) (Figure 1A). Decontamination efficiency was different between DNA and cDNA samples for particular contaminant strains. *M. luteus* was significantly less efficiently removed for cDNA samples compared to *E. coli* (*p* = 0.024), but not compared to *P*. fluorescens (*p* = 0.999) (Figure 1B). The decontamination efficiency of *P*. fluorescens was similar (*p* = 0.057) for cDNA samples compared to DNA samples, whereas *E. coli* (*p* = 0.031) and *M. luteus* (*p* = 0.011) differed for both sample types.

### 3.2. Decontamination Strategy Shifted Bacterial Contamination Diversity

A total of 3,005,661 and 1,389,711 sequences were obtained for DNA or cDNA samples, which corresponded to 2699 or 2256 bacterial OTUs, respectively. 1756 bacterial OTUs were found in both sample types, while 943 and 500 OTUs were solely present for DNA or cDNA samples, respectively. The ACE, shannon index as well as evenness were significantly different for DNA (*p* = 0.001, *p* = 1.0 × 10^−6^, *p* = 3.0 × 10^−9^) as well as for cDNA samples (*p* = 0.002, *p* = 1.2 × 10^−4^, *p* = 9.1 × 10^−10^) between the decontamination methods (DKAs) (Table 3). The OTU richness of respective DKA was significantly different for DNA samples (*p* = 6.8 × 10^−5^), but not for cDNA samples (*p* = 0.124). In turn, chao1 was not significantly different for DNA samples (*p* = 0.102), but for cDNA samples (*p* = 0.001).

### 3.3. Effect of Decontamination Treatment on Tick-Borne Microbiome

The composition of the tick-borne microbiome was highly impacted by respective decontamination treatment for DNA and cDNA samples (see Figure 2). While the composition of the microbiome after DKA 1, DKA 2 and DKA 4 was more similar to the positive control, the DKA 3 (5% sodium hypochlorite) clustered closer to the negative controls. These results were found for DNA as well as for cDNA samples (compare Figure 2A with Figure 2B).

The Euclidean distances revealed five bacterial clusters for DNA and cDNA samples, which were similarly organized in clusters 1 to 3 (Figure 3). However, negative control (DKA 3 NC, DKA 1 NC and DKA 4 NC) clustered differently between DNA and cDNA samples. Bacterial community compositions retrieved from ticks after 5% sodium hypochlorite treatment (DKA 3) clustered individually, whereas ticks after DKA 1, DKA 2 and DKA 4 treatment clustered with PC (cluster 2) (Figure 3A).

Bacterial community compositions of ticks after 5% sodium hypochlorite treatment (DKA 3) had high abundances of bacterial genera that were not part of the contamination solution (36.1% for DNA samples and 52.3% for cDNA samples) (Figure 3 and Table 4). In turn, the bacterial community compositions of ticks after DKA 1, DKA 2 or DKA 4 treatment and positive control were mostly similar to each other and had a low abundance of non-contaminants (Figure 3 and Table 4). Bacterial community compositions of non-treated but contaminated ticks (PC) were characterized by low abundances of non-contaminants (3.5% for DNA samples and 3.7% for cDNA samples). In comparison, bacterial community compositions of non-contaminated but decontaminated ticks (NC) consisted of high abundances of non-contaminants (83.9% ± 7.0% for DNA samples, and 97.25% ± 2.5% for cDNA samples) (Figure 3 and Table 4). For detailed analyses, which treatment and replicate clustered with each other, see also Appendix A.

## 4. Discussion

### 4.1. Tick Decontamination by 5% Sodium Hypochlorite Treatment Was More Efficient Compared to Other Decontamination Treatments

The majority of previous studies assessed the internal tick microbiome without any decontamination strategy of the external tick microbiome. Studies with a decontamination strategy mainly applied ethanol-based decontamination [26,36,59], methodologically similar to our approach (DKA 1). However, our decontamination efficiency tests revealed that 5% sodium hypochlorite was the most efficient agent for tick surface decontamination followed by DNA Away, RSDL, and finally by 70% ethanol (Figure 2 and Figure 3). Sodium hypochlorite was mainly used to eliminate DNA contaminants and dental infections and for laboratory surface decontamination [27,60]. Our literature research revealed that only one study used bleach solutions to decontaminate the external tick microbiome [22], while two other studies combined bleach and ethanol [23,24]. However, these studies did not address the decontamination efficiency of bleach solutions or compare different decontamination approaches but do reveal meaningful conclusions. Sodium hypochlorite is known to randomly disrupt cellular metabolic processes resulting in the degeneration of phospholipids. This causes oxidative reactions with irreversible enzymatic inactivation forming cytoplasmic chloramines that inhibit cellular metabolism finally leading to a degradation of lipid and fatty acids [41,61]. If 5% sodium hypochlorite was used as the primary reactive substance in the decontamination solution, a degradation of nucleic acids is expected, which can cause a lower DNA and RNA content in the external tick-microbiome.

### 4.2. Gram-Positive Contaminants Were Removed Less Efficiently Compared to Gram-Negative Contaminants

The highest quantitative removal of *E. coli*, *P. fluorescens* and *M. luteus* as contaminants was achieved by a 5% sodium hypochlorite treatment (Figure 1), which is in line with previous results from biofilm removal [60]. The removal efficiency was similar to DNA and cDNA samples for all tested contaminants (Figure 1). However, the decontamination of Gram-positive *M. luteus* was significantly less efficient compared to other contaminants, which is in line with previous sodium hypochlorite treatments [62]. Moreover, Gram-positive bacteria and their nucleic acids were more challenging to eliminate and bacteria withstood sodium hypochlorite treatments of different concentrations, including 5% sodium hypochlorite and an incubation period from 1 to 10 min [62,63,64]. Besides, other decontamination treatments (DKA 1, DKA 2 and DKA 4) were also less efficient in removing the contaminant strain *M. luteus* compared to the removal of the Gram-negative contaminants, such as *E. coli* and *P. fluorescens*. However, a direct comparison of gene copy numbers and sequence reads are not appropriate to estimate the removal efficiency, as different primer sets and PCR conditions were employed and these methodological implications were addressed previously [65].

### 4.3. Effect of 5% Sodium Hypochlorite Treatment on Tick-Borne Microbiome

Ticks treated with 5% sodium hypochlorite revealed lower richness compared to other DKAs, as the number of singletons and doubletons was much lower (see Appendix A). Therefore, 5% sodium hypochlorite treatment reduced the number of rare sequences and the overall read coverage compared to the other DKAs (Table 3). In turn, 5% sodium hypochlorite treated tick microbiomes had a more even distribution of sequences as the number of rare sequences of other DKAs caused uneven distributions. The effect of sodium hypochlorite on amplicon sequencing results can be based on (i) a higher degradation of free and less abundant nucleic acid sequences, (ii) the Illumina-based synthesis reaction during sequencing or (iii) or Illumina-based dye reactions. Interestingly, sodium hypochlorite concentrations between 0.9 to 6% (vol./vol.) caused no negative effect on the PCR amplification process [66,67]. The MiSeq system guide of Illumina routinely recommend a solution containing 1.25% (vol./vol.) sodium hypochlorite in a washing step after post-run to eliminate contaminations of previous sequencing runs [68]. Furthermore, the manufacturer recommends being careful in this washing step since high sodium hypochlorite concentrations lead to failures in cluster generation in subsequent runs [68]. Therefore, a carryover of even trace amounts of sodium hypochlorite into the sequence reactions can have a high effect on sequencing results. However, in our experiments, we included in each DKA a strict regime to remove DKA components by (i) discarding the supernatant after DKA treatment, (ii) evaporating the remaining solution in SpeedVac and (iii) washing nucleic acid extracts three times with RNAse and DNAse free water. After amplification, amplicons were desalted before sequence reactions, which removed potential traces of sodium hypochlorite or other DKA compounds.

Microbiomes retrieved from ticks that were decontaminated with 5% sodium hypochlorite clustered together with NCs (Figure 2). However, some NCs contained very low sequence read amounts of OTUs affiliated to *P. fluorescens* and *E. coli* (Table 4), which were similar to the sequences of our contamination solution (Table 2). Likewise, the same sequences were also found as part of the tick microbiome in previous studies without any contamination [34,69], indicating that these OTUs are part of the indigenous tick microbiome. Variation in the bacterial community composition of the tick microbiome has been frequently found [70]; therefore, standardised microbiomes from the NCs were not expected and also revealed a high variance (Figure 2). The microbiomes of ticks decontaminated with DNA Away, RSDL or 70% ethanol were highly biased by sequences of the contamination solution and clustered with the PCs (Figure 3).

## 5. Conclusions

The decontamination of ticks is commonly carried out with ethanol, which was the most inefficient agent in our study. Strategic rethinking in the decontamination of ticks is needed as sodium hypochlorite treatment was superior to other DKAs. Although sodium hypochlorite may negatively affect sequencing results, our findings suggest that no loss in frequent sequences or shifted community composition compared to the NCs occurred. In this study, we focused on adult *I. ricinus* individuals as decontamination targets and upcoming approaches should address the transferability of our methodology to decontaminate other plant or animal targets (including larvae or nymphs of *I. ricinus* or other *Ixodes* species).

## Figures and Tables

**Figure 1 microorganisms-08-00987-f001:**
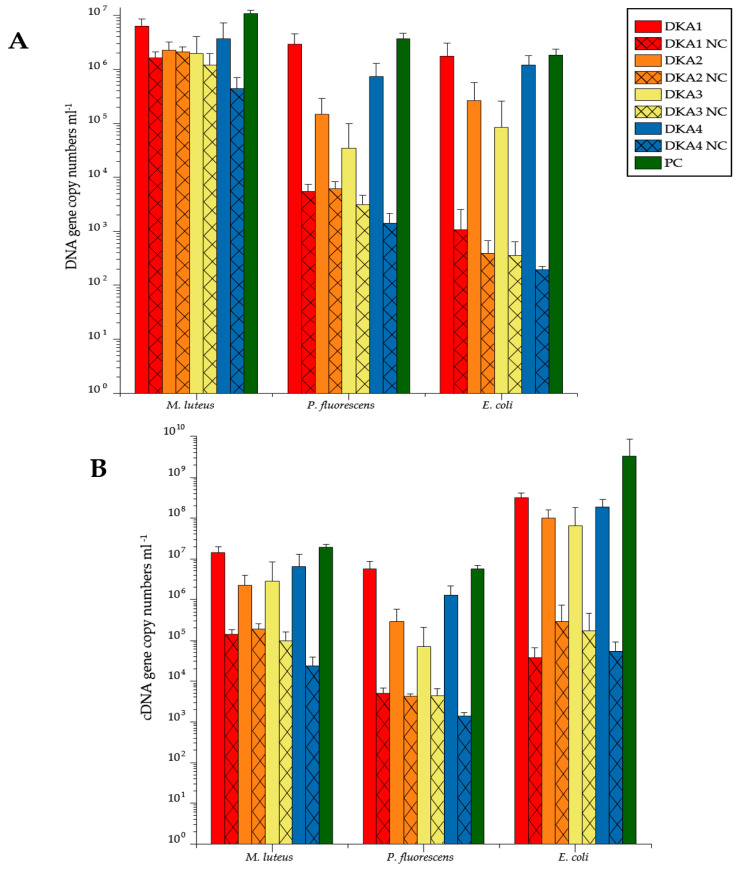
Reduction of artificial external tick-microbiome contaminants after decontamination with 70% ethanol (DKA 1), DNA Away (DKA 2), 5% sodium hypochlorite (DKA 3) and RSDL (DKA 4); positive control (PC), negative controls (NC) for DNA (**A**) and cDNA samples (**B**). For colors and patterns, see figure legend. Error bars indicate a standard error: *n* = 10 (each treatment, PC) or *n* = 3 (NC). The details of treatments and replicates are summarized in Table 1.

**Figure 2 microorganisms-08-00987-f002:**
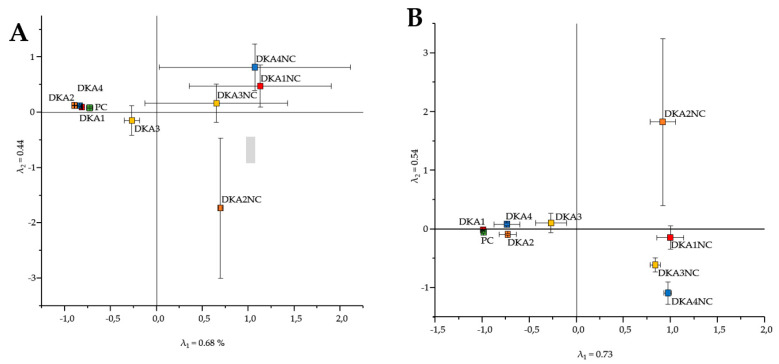
Correspondence analysis of bacterial community compositions for DNA (**A**) and cDNA samples (**B**) of 70% ethanol (DKA1), DNA Away (DKA 2), 5% sodium hypochlorite (DKA 3) and RSDL (DKA 4) decontaminated ticks. PC (positive control) without decontamination and NC (negative control) ticks without contamination. The error bars represent the SD of *n* = 10 (each treatment, PC) or *n* = 3 (NC). The details of treatments and replicates are summarized in Table 1.

**Figure 3 microorganisms-08-00987-f003:**
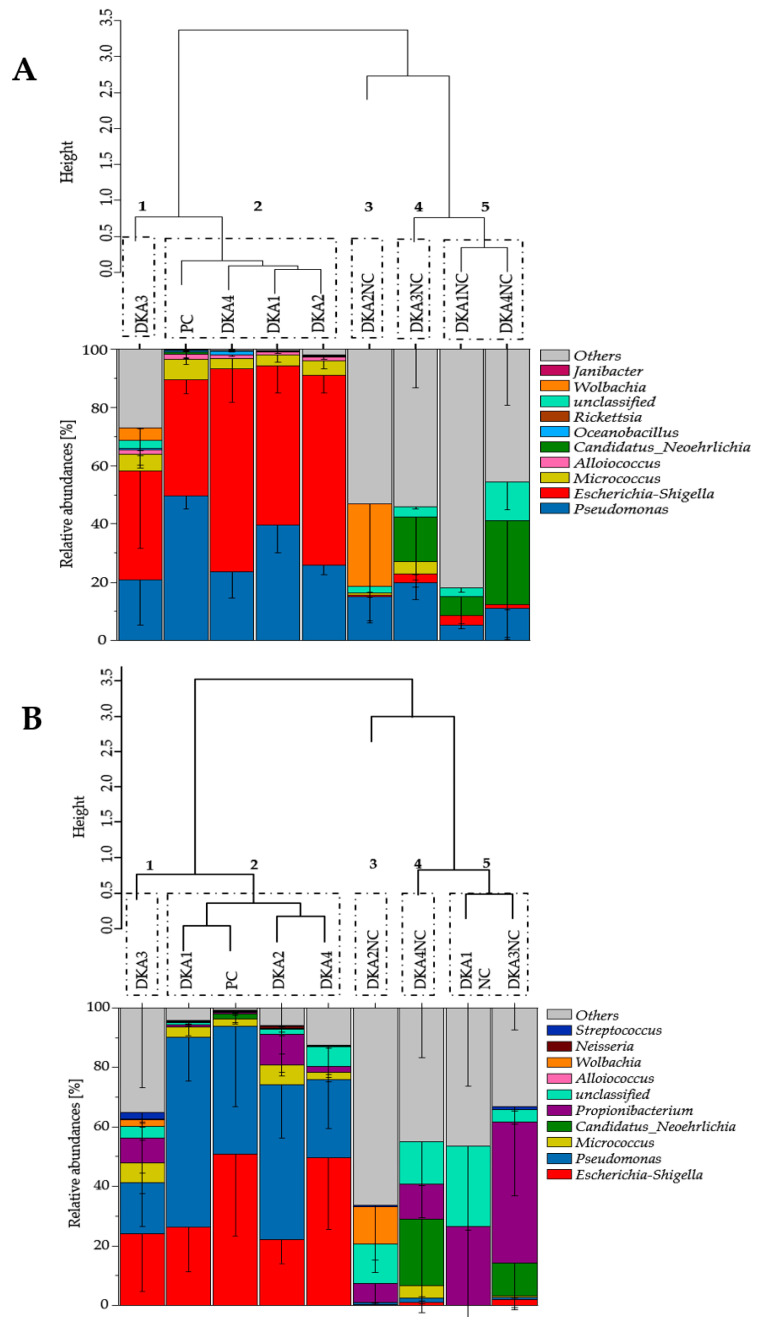
Euclidean distance matrix based on the ward.D2 method (above) and bacterial community composition on genus level (below) for the DNA (**A**) and cDNA samples (**B**) of 70% ethanol (DKA 1), DNA Away (DKA 2), 5% sodium hypochlorite (DKA 3) and RSDL (DKA 4) decontaminated ticks. PC (positive control) without decontamination and NC (negative control) ticks without contamination. For colors and patterns, see figure legend. The bacterial community composition of the ten most abundant genera are denoted and other genera are summarized as “others”, and composition of each sample can be in Appendix A. For clusters, a height of 0.5 was chosen, numbered and denoted in dashed boxes. The details of treatments and replicates are summarized in Table 1.

**Table 1 microorganisms-08-00987-t001:** Decontamination strategies to assess the efficiency of four decontamination solutions.

Decontamination Solution (5 min)	Abbreviation	Contamination	Number of Independent Ticks
**70% ethanol**	DKA 1	yes	10
**DNA Away**	DKA 2	yes	10
**5% sodium hypochlorite**	DKA 3	yes	10
Reactive Skin Decontamination Lotion (**RSDL**)	DKA 4	yes	10
**70% ethanol**	DKA 1 NC	no	3
**DNA Away**	DKA 2 NC	no	3
**5% Sodium hypochlorite**	DKA 3 NC	no	3
**RSDL**	DKA 4 NC	no	3
**No**	PC	yes	10

**Table 2 microorganisms-08-00987-t002:** Composition of defined contamination solution.

Contaminants	Cell Number mL^−1^	Volume of Solution (in Total)	Percentage Volume in Solution per Tick
***Micrococcus luteus***	1.1 × 10^9^	650 µL	28.9%
***Pseudomonas fluorescens***	1.4 × 10^9^	650 µL	28.9%
***Escherichia coli***	6.5 × 10^9^	650 µL	28.9%
**Human sweat**	not determined	200 µL	8.9%
**Dog salvia**	not determined	100 µL	4.4%

**Table 3 microorganisms-08-00987-t003:** OTU diversity indices from the tick microbiome after decontamination treatment with 70% ethanol (DKA 1), DNA Away (DKA 2), 5% sodium hypochlorite (DKA 3) and RSDL (DKA 4) for DNA (A) and cDNA samples (B). OTUs ≥ 1% of relative abundances were included. Mean values are indicated; *n* = 10 or *n* = 3. The details of treatments and replicates are summarized in Table 1.

**A**	**DNA**	**OTU Richness**	**Shannon**	**Pielou’s Evenness**	**S.chao1 ***	**S.ACE ****
	**DKA 1**	86 ± 17	0.9 ± 0.1	0.2 ± 0.0	359 ± 95	371 ± 118
	**DKA 2**	101 ± 50	1.0 ± 0.2	0.2 ± 0.1	1229 ± 1750	697 ± 599
	**DKA 3**	43 ± 22	1.6 ± 0.5	0.5 ± 0.2	136 ± 147	124 ± 122
	**DKA 4**	99 ± 65	0.8 ± 0.2	0.2 ± 0.1	536 ± 563	384 ± 378
	**DKA 1 NC**	35 ± 8	1.8 ± 0.2	0.5 ± 0.1	44 ± 16	41 ± 12
	**DKA 2 NC**	22 ± 4	1.4 ± 0.5	0.5 ± 0.2	26 ± 6	27 ± 6
	**DKA 3 NC**	17 ± 1	1.8 ± 0.2	0.6 ± 0.1	21 ± 3	21 ± 3
	**DKA 4 NC**	29 ± 3	1.9 ± 0.9	0.6 ± 0.3	42 ± 8	41 ± 6
	**PC**	112 ± 31	1.1 ± 0.1	0.2 ± 0.0	530 ± 270	512 ± 205
**B**	**cDNA**	**OTU Richness**	**Shannon**	**Pielou’s Evenness**	**S.chao1 ***	**S.ACE ****
	**DKA 1**	101 ± 60	1.0 ± 0.3	0.2 ± 0.1	370 ± 298	389 ± 348
	**DKA 2**	31 ± 24	1.3 ± 0.2	0.5 ± 0.2	44 ± 34	46 ± 32
	**DKA 3**	44 ± 24	2.0 ± 0.5	0.6 ± 0.1	88 ± 65	78 ± 52
	**DKA 4**	104 ± 144	1.6 ± 1.0	0.4 ± 0.1	168 ± 239	145 ± 194
	**DKA 1 NC**	12 ± 6	1.7 ± 0.6	0.8 ± 0.2	20 ± 10	28 ± 7
	**DKA 2 NC**	60 ± 29	2.7 ± 0.7	0.7 ± 0.1	75 ± 31	77 ± 32
	**DKA 3 NC**	20 ± 12	1.8 ± 0.6	0.6 ± 0.1	27 ± 10	33 ± 6
	**DKA 4 NC**	20 ± 8	1.9 ± 0.8	0.6 ± 0.2	34 ± 3	39 ± 2
	**PC**	92 ± 49	0.8 ± 0.3	0.2 ± 0.1	300 ± 162	313 ± 162

* Bias-Corrected Chao1, ** Abundance-Based Coverage estimator.

**Table 4 microorganisms-08-00987-t004:** Relative Operational Taxonomic Unit (OTU) abundances from the tick microbiome assigned to *M. luteus*, *P. fluorescens* and *E. coli* after decontamination treatment with 70% ethanol (DKA 1), DNA Away (DKA 2), 5% sodium hypochlorite (DKA 3) and RSDL (DKA 4) for DNA (A) and cDNA samples (B). Mean values are indicated; n = 10 or n = 3. Details of treatments and replicates are summarized in Table 1. Relative OTU abundances were calculated for each tick and summarized decontamination strategies by the OTU count table for DNA or cDNA samples (Appendix A).

A	DNA	Relative Abundance of Sequences of Bacterial Contaminants [%]			B	cDNA	Relative Abundance of Sequences of Bacterial Contaminants [%]		
		*M. luteus*	*P. fluorescens*	*E. coli*			*M. luteus*	*P. fluorescens*	*E. coli*
	**DKA 1**	3.6	39.7	54.6		**DKA 1**	3.4	64.1	26.2
	**DKA 2**	5.1	25.8	65.2		**DKA 2**	6.8	52.2	22.0
	**DKA 3**	5.7	20.8	37.4		**DKA 3**	6.6	17.1	24.0
	**DKA 4**	3.5	23.5	69.7		**DKA 4**	2.3	26.4	49.6
	**DKA 1 NC**	0.1	5.3	3.1		**DKA 1 NC**	0.0	0.0	0.0
	**DKA 2 NC**	0.7	15.1	0.6		**DKA 2 NC**	0.1	0.8	0.2
	**DKA 3 NC**	4.4	19.7	3.0		**DKA 3 NC**	0.5	0.6	2.1
	**DKA 4 NC**	0.0	11.0	1.2		**DKA 4 NC**	4.3	1.3	1.1
	**PC**	7.0	49.8	39.7		**PC**	2.4	43.1	50.8

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
