# Peer review of "Analysis of Tick Surface Decontamination Methods"

_microorganisms, 2020, doi:10.3390/microorganisms8070987_

Round 1

Reviewer 1 Report

This manuscript addresses a timely issue, that of appropriate "cleaning" of ticks prior to microbiome analysis.  However, rather than simply subjecting a group of ticks to "cleaning" and comparing to a group that had not been "cleaned", the authors chose to introduce contaminants to the tick surface.  

There is no assessment of whether dipping ticks in a cocktail of bacteria (and other fluids) actually serves as an appropriate method to introduce these bacteria to the ticks.  After placing ticks in the bacterial solutions for  5 minutes, the authors, decontaminated the ticks for 5 min.  No mention is made as to whether they used vigorous shaking (an important factor).

The data show that ticks that were not subjected to contamination with the bacteria have significant amounts of the bacteria.  The authors state that the bacteria were chosen because they are not associated with ticks... so where did the bacteria come from in the "noncontaminated" ticks?

The authors also used dog saliva and human sweat as complex mixtures to assess contamination, but this is never analyzed.  The authors crunch up the whole tick to make nucleic acids for their preps, so in their microbiome analysis they are assessing the internal as well as the external microbiome.  This further complicates the analysis, and assessment of which cleaning method worked the best.

The manner in which the authors presented their data obscures huge variation.

I do not understand why the authors chose to use different group sizes for the contaminated and non-contaminated tick groups (10 vs 3)... surely this makes comparisons more complicated.

The authors state that the microbiome data indicated that there was very little of the spiked bacteria detected, yet the qPCR indicated that there was quite a bit... this doesn't add up.  Can they explain this inconsistency?

Line 89-90 - When each tick is transferred with a disinfected tweezer… was the tweezer disinfected between each tick? How long were ticks stored at 4 degrees?

Line 108 – what is the purpose of this????

The end result of the paper seems to be valid (common sense), yet there are many issues with the paper itself:  The English is NOT of an appropriate standard and needs to be reviewed by a native English speaker.  There are many awkward sentences and nonsensical sentences.  The low and disparate sample numbers are problematic.  The experimental design seems overcomplicated, and then not thoroughly addressed.

A few examples of English issues:

Page 1 line 34: awkward sentence… TBP are the reason for….

Line 37 – broadened

Line 39 – tense

Lines40-42: nonsensical sentence… talking about variable regions of the 16S rRNA gene used for analysis; and then say … “were named as microbiome”   ??

Line 79 – RNA is not transcribed to copy DNA

Line 83 – poor grammar

Line 270 – Level is inappropriate term here.

Author Response

Comments from the editors and reviewers:

We are submitting our revision for our manuscript. We appreciate the time the editor and the reviewers put into improving our submission. We have systematically addressed each reviewer`s comments and suggestions, as detailed below, and we have made changes and improvements throughout the manuscript. In addition, we have edited our English to improve the readability. All citations named in this rebuttal are included in the revised manuscript version or were separately listed in the rebuttal.

Reviewer 1

This manuscript addresses a timely issue, that of appropriate "cleaning" of ticks prior to microbiome analysis.  However, rather than simply subjecting a group of ticks to "cleaning" and comparing to a group that had not been "cleaned", the authors chose to introduce contaminants to the tick surface.

Reply: We thank the reviewer for their comment. The bacterial composition of the tick microbiome is highly variable and influenced by its surrounding environment and the host environment, which has been shown by many studies and was recently summarized in a review by Greay et al. 2018. Likewise the external microbiome will be also highly variable and hard to compare between individual ticks. The focus of our study was the quantitative assessment of four decontamination approaches, which can only be quantified when known and consistent contaminants can be addressed. Therefore, we used the same mixture of bacterial contaminants to evaluate each decontamination method in detail. In addition, our negative controls (DKA1-4 NC) were not contaminated but decontaminated. These negative controls revealed a highly variable community composition in the microbiome on DNA and cDNA level (Figure 2), and therefore the evaluation of decontamination efficiency of each DKA without prior contamination is much more difficult. Therefore, we believe the assessment of the decontamination of ticks inoculated in a consistent contamination mixtures was appropriate.

There is no assessment of whether dipping ticks in a cocktail of bacteria (and other fluids) actually serves as an appropriate method to introduce these bacteria to the ticks.  After placing ticks in the bacterial solutions for 5 minutes, the authors, decontaminated the ticks for 5 min.  No mention is made as to whether they used vigorous shaking (an important factor).

Reply: Thank you very much for showing this unclear part. The contamination procedure for tick surfaces are from our literature search novel and we deduced a procedure of contamination of human skin (Ballantyne et al. (2015). To clarify our contamination procedure we included this information with a citation in the introduction.

In addition, the procedures of decontamination are heterogeneously carried out in the literature. Therefore, a standardized protocol of decontamination are also not available. The time frame of decontamination procedures was rarely reported and only Kemp et al. (2005) explained the decontamination by sodium hypochlorite in detail including time frames. Kemp et al. (2005) used 5 min contact time of sodium hypochlorite on bones and teeth surface without shaking. To ease the comparability with this study and the lack of public available decontamination time frames from other studies, we decided also to decontaminate in 5 min without shaking. To address, the lack of present information of decontamination procedures, we added this information in the introduction of the revised manuscript version.

The data show that ticks that were not subjected to contamination with the bacteria have significant amounts of the bacteria. The authors state that the bacteria were chosen because they are not associated with ticks... so where did the bacteria come from in the "non-contaminated" ticks?

Reply: Thank you very much for highlighting this critical point. During our investigation for suitable contaminants, we examined many papers that conducted research on tick microbiomes by 16S rRNA gene sequencing. We focused on species of genera MicrococcusPseudomonas and Escherichia as part of the contamination mixture as these genera were rarely found in tick microbiome and were used to assess the nucleic acid extraction procedures from soil (as summarized in Bakken & Frostegård, 2006). In addition, OkÅ‚a et al. (2012) and Matsuo et al. (2004) stated that members of the genera Escherichia and Micrococcus were never found in tick microbiomes. In addition, members of these genera were even used as artificially added bacterial species for tick experiments from these authors.

In addition, Binetruy et al. (2019) detected low number of sequence reads from the genus Pseudomonas (whole tick and external microbiome), and therefore high quantities of members of this genus were not expected. Therefore, we consider a low probability of the presence of sequence motifs of these species. However, ticks can be inoculated by a broad bacterial diversity in nature and it is hard to fully exclude the presence of bacterial species. Our sequence read numbers from OTUs affiliated to these genera from the negative controls were also very low. However, we used a different primer set to quantify sequence motifs of MicrococcusPseudomonas or Escherichia, and each primer set and PCR condition may also differ between each other, and has its own pitfalls as explained earlier by von Wintzingerode et al. (1997). To address the problem of comparability between these data sets, we stated in the revised version of the discussion that a direct comparison of gene copy numbers and sequence reads were not appropriate.    

The authors also used dog saliva and human sweat as complex mixtures to assess contamination, but this is never analyzed.  The authors crunch up the whole tick to make nucleic acids for their preps, so in their microbiome analysis they are assessing the internal as well as the external microbiome.  This further complicates the analysis, and assessment of which cleaning method worked the best.

Reply: Thank you for your summary and the missing information. We assessed the microbiome from dog saliva and human sweat in a literature research and revealed that these microbiomes are also heterogeneous (Ruparell et al. (2020) and Okamoto et al. (2018)). To show the diversity in these microbiomes and the overlap to the sequence diversity in our microbiome analyses, we added a new table in the supplementary part of the revised manuscript. As the sequence overlap is low and the focus of the study was different, we think that this information is best in the supplementary part.   

Furthermore, we know that it would be a more natural way to analyze the external and internal microbiome separately for testing the decontamination efficiency, like already shown in Binetruy et al. (2019). However, the efficiency of bacterial removal from tick surfaces has been done –to our knowledge- only by Binetruy et al. (2019) and the probability to obtain the entire external microbiome is very crucial for a quantitative assessment of external contaminants. As the quantitative approach on DNA and cDNA level was our focus, we believe that the assessment of the entire tick microbiome has a higher recovery rate from our contamination solution than a separation approach of external and internal tick microbiome approach.

Okamoto, H.; Koizumi, S.; Shimizu, H.; Cho, O.; Sugita, T. Characterization of the axillary micobiota of japanese male subjects with spicy and milky odor types by pyrosequencing. Biocontrol Science 2018, Vol.23, No. 1, 1-5, doi:10.4265/bio.23.1

The manner in which the authors presented their data obscures huge variation.

Reply: The main message in the main document summarizes the principal findings. For readers with high interest can visit our data sets in the supplementary part, which summarize the variation in the sequence reads of each OTU and many other additional information. We believe that our clear-cut focus in the main document should not be changed.

I do not understand why the authors chose to use different group sizes for the contaminated and non-contaminated tick groups (10 vs 3)... surely this makes comparisons more complicated.

Reply: Thank you for your request. The focus of our study was the test of the efficacy of the decontamination methods and therefore we increased the replicates amount to ten. As the DNA and cDNA level was second focus on this study we do not reduced the amount of replicates. In addition, we sampled all ticks from the same site and the same date and we assumed that the non-contaminated tick microbiome was more similar to each other than the contaminated tick microbiome. Therefore, we reduced the amount of negative controls compared to our main goals. Many studies have a replicate number of three and therefore we were beforehand confident that three replicates should be enough to cover the variability in the tick microbiome. From our point of view, the number of replicates are sufficient to meet the main goals of this study.

The authors state that the microbiome data indicated that there was very little of the spiked bacteria detected, yet the qPCR indicated that there was quite a bit... this doesn't add up.  Can they explain this inconsistency?

Reply: Thank you very much for highlighting this critical point. A direct comparison between results of different primer sets and PCR conditions for quantitative PCR and for PCR and subsequent amplicon sequencing is crucial. As already stated above, we addressed this problem in the revised version of the discussion. In addition, the preparation of sequence libraries, the sequencing procedure by Illumina technology and sequence data processing are additional pitfalls beside the drawbacks of PCR. Therefore, a direct comparison between both data sets and inbetween qPCR data sets are misleading and was not our intention in the manuscript. We changed the manuscript throughout the manuscript to remove such direct comparisons.

Line 89-90 - When each tick is transferred with a disinfected tweezer… was the tweezer disinfected between each tick? How long were ticks stored at 4 degrees?

Reply: We are sorry that this information was not presented clear enough. Of course we disinfected the tweezer for each tick and ticks were stored after sampling for 16 hours. We added in the revised version in the section “Tick Sample Collection” this additional information.

Line 108 – what is the purpose of this????

Reply: Hopefully we address your request correctly for this line. We have introduced the decontamination methods twice: once in the introduction and twice in the material & method section by citations. In the revised version of the introduction we have inserted the citations of the previous M&M section and we removed these citations in the M&M section.  

The end result of the paper seems to be valid (common sense), yet there are many issues with the paper itself:  The English is NOT of an appropriate standard and needs to be reviewed by a native English speaker.  There are many awkward sentences and nonsensical sentences.  The low and disparate sample numbers are problematic.  The experimental design seems overcomplicated, and then not thoroughly addressed.

Reply: Thank you very much for giving us such an excellent critical review. However, we are glad that you think that our results our sound. We have addressed all your comments and think that the overall information are now presented clearer for the readers. We also checked the English language and hopefully our English now fit the journal standard.

A few examples of English issues:

Page 1 line 34: awkward sentence… TBP are the reason for….

Reply: We are sorry for this inept start of this sentence. We rephrased this and the following sentence in the revised version.

Ixodes ricinus is the most common tick species in Germany and vector of a vast range of TBP, including BorreliaRickettsia or Coxiella that can cause Lyme disease, rickettsial disease or Q fever.”

Line 37 – broadened

Reply: Thanks. We exchanged to “enhanced”.

Line 39 – tense

Reply: We changed the tense to present perfect.

Lines 40-42: nonsensical sentence… talking about variable regions of the 16S rRNA gene used

for analysis; and then say … “were named as microbiome”   ??

Reply: Thank you for showing this inappropriate sentence. We deleted this part of the sentence.

Line 79 – RNA is not transcribed to copy DNA

Reply: Thank you for addressing this mistake. At first it is not the copy but complementary DNA. Furthermore, we added “reverse” to make clear that RNA is reverse transcribed to cDNA. We replaced this information in the revised manuscript version.

Line 83 – poor grammar

Reply: Thank you for showing our inconvenience sentence structure, which was rephrased in the revised manuscript version.

“Permission for tick collection at the study site was requested at the government of Lower Franconia at Würzburg, workspace animal protection and animal testing. For this examination, no formal permit was required.”

Line 270 – Level is inappropriate term here.

Reply: Level is a frequently used term in literature especially if different kinds of nucleic acids been analyzed. Similarly, we also addressed in our manuscript two different types of nucleic acids (DNA and RNA/cDNA). As this wording is common we will remain level in the revised manuscript version.

Reviewer 2 Report

Just general remarks:

A microbiome is a community of commensal, symbiotic, and pathogenic microorganisms that naturally reside in the organism. None of the bacteria from the current study is part of tick microbiome as stated by authors.

Is this study really about tick microbiome? Is the title appropriate?

“Decontamination of external microbiome from ticks is essential to retrieve unbiased internal tick-borne microbiome and to unambiguously detect its pathogens.” I think that the last statement from the abstract and the paper title cannot be concluded from the current study itself. The authors decided to assess decontamination efficacy of artificial contamination with bacteria rather than microbiome.

The study does not address viruses and their removal what would be interesting. The authors mention TBE as an important tick-borne pathogen.

Author Response

Reviewer 2

Just general remarks:

A microbiome is a community of commensal, symbiotic, and pathogenic microorganisms that naturally reside in the organism. None of the bacteria from the current study is part of tick microbiome as stated by authors.

Is this study really about tick microbiome? Is the title appropriate?

“Decontamination of external microbiome from ticks is essential to retrieve unbiased internal tick-borne microbiome and to unambiguously detect its pathogens.” I think that the last statement from the abstract and the paper title cannot be concluded from the current study itself. The authors decided to assess decontamination efficacy of artificial contamination with bacteria rather than microbiome.

Reply: Thank you for addressing this critical point, and we agree. First, we changed the title to “Analysis of tick surface decontamination methods” and further we changed the last part of abstract to “Therefore, efficiency of decontamination was best by 5% sodium hypochlorite and is recommended for upcoming studies to address unbiased detection of tick-borne pathogens.” in the revised manuscript version.

The study does not address viruses and their removal what would be interesting. The authors mention TBE as an important tick-borne pathogen.

Reply: Thank you very much for your comment. It is well known that viruses, such as TBEV, represent an important research object concerning tick-borne pathogens, and so we decided to mention one of these important viruses. Nevertheless, the focus of this study a bacterial tick-borne pathogens. To avoid confusions by the readers, we exchanged TBE with Coxiella in the revised manuscript version. We hope that other researcher will address the virus based decontamination of tick surfaces.

Round 2

Reviewer 1 Report

Really... you should get the advice of a native English speaker!

Line 40 change "were" to "have been"... "were" makes it sound like you did it in this paper.

Line 132: change to "The host skin microbiome (such as that of mammals like humans or dogs)..."

Line 140 ff: surfaces should be singular... we don't speak of tick skin... tick's don't have skin.

Line 352: Level is NOT THE CORRECT TERM.... "denaturation at 95 °C for 3 min for DNA samples or 30 sec for cDNA samples followed..."

You are not looking at sequence motifs... you are looking at sequences.

Figure 3 is upside-down.... surely you could put panel A above panel B in a stacked arrangement... it is very awkward to present figures in landscape arrangement.

Author Response

Comments from the reviewer 1:

We thank the reviewer 1 for the constructive comments and criticism of our manuscript again, which helped us to improve the manuscript.

Really... you should get the advice of a native English speaker!

Reply: So we sent the manuscript to a native English speaker and are hopeful that the English is now of an appropriate standard. All changes can be revisited in the revised version of the manuscript.

Line 40 change "were" to "have been"... "were" makes it sound like you did it in this paper.

Reply: Thank you for highlighting this mistake. We changed “were” to “have been” in the revised version of the manuscript.

Line 132: change to "The host skin microbiome (such as that of mammals like humans or dogs)..."

Reply: We filled in “such” in the revised version of the manuscript.

Line 140 ff: surfaces should be singular... we don't speak of tick skin... tick's don't have skin.

Reply: Thank you for pointing out this mistake. We changed “tick skin” to “tick surface” on lines 63, 67 and 70 in the revised version of the manuscript.

Line 352: Level is NOT THE CORRECT TERM.... "denaturation at 95 °C for 3 min for DNA samples or 30 sec for cDNA samples followed..."

Reply: We changed “on DNA/cDNA level” to “for DNA/cDNA samples” at each part of the revised version of the manuscript as well as on the supplementary part (Figures S1 and S2; Tables S1, S2 and S3).

You are not looking at sequence motifs... you are looking at sequences.

Reply: We deleted “motifs” and changed “sequence” to “sequences” at each part of the revised version of the manuscript

Figure 3 is upside-down.... surely you could put panel A above panel B in a stacked arrangement... it is very awkward to present figures in landscape arrangement.

Reply: We changed Fig. 3A/B from landscape to portrait in the revised version of the manuscript.